# MicroRNA-206 suppresses growth and metastasis of breast cancer stem cells via blocking EVI-1-mediated CALR expression

**Dapeng Sun[1]\*, Chenguang Li[1], Fengxiang Zhang[2]\***

**1** Department of Clinical Pharmaceutics, The First Affiliated Hospital of JINZHOU Medical University, Jinzhou, China, **2** Department of Critical Care Medicine, The First Affiliated Hospital of JINZHOU Medical University, Jinzhou, China

\* zhangfengxiang64@163.com (FZ); canghaiguanri@126.com (DS)

## Abstract

Aim to investigate the effect of miR-206 on the growth and metastasis of breast cancer stem cells and clarify the precise mechanism of miR-206 on EVI-1-mediated CALR expression in driving malignant phenotype. Our results showed that miR-206 mimics suppressed CALR expression, inhibited the proliferation and metastasis ability of breast cancer stem cells and finally induced cellular apoptosis. Over-expression of CALR could effectively attenuate the cytotoxic effect of miR-206. Further studies demonstrated that EVI-1 could be served as a key regulator of miR206-mediated CALR expression. Elevation of EVI-1 can reverse the function of miR-206 on induction of CALR.

**Data Availability Statement:** All relevant data are within the paper and its Supporting Information files.

**Funding:** This study was supported by Liaoning Provincial Natural Science Foundation

## Introduction

Breast cancer is the most common malignant tumor in female cancers. Currently, the main treatment methods are surgery, radiotherapy, chemotherapy and biological therapy. However, the development of drug resistant is an obstacle to successfully treatment for breast cancer. Recently studies have confirmed that there are a small subgroup of breast cancer stem cells in breast cancer tissues [1, 2], which not only have high tumorigenicity and high invasion and metastasis, but also have the characteristics of resistance to chemotherapy and radiotherapy. They play important roles in the development, metastasis and resistance in breast cancer. Therefore, discovering novel biological markers of breast cancer stem cells, developing corresponding molecular targeted drugs, and carrying out multi-sites targeted drug therapy will contribute to effective treatment for breast cancer [3, 4].

Calreticulin (CALR), a multifunctional calcium-binding molecular chaperone, mainly locate in cellular endoplasmic reticulum (ER). The abnormal expression of CALR occurs in different types of tumors [5]. Previous studies showed that calreticulin could significantly affect cell proliferation, invasion, induction of apoptosis and angiogenesis in glioma tissues [6]. Studies have confirmed that calreticulin expression was up-regulated and mainly localized in the cytoplasm in breast cancer MDA-MB-231 stem cells. The expression level of calreticulin

(20180550601&2019-ZD-0833); Liaoning Provincial Ediucation Foundation (JYTCZR2020051). The funders had no role in study design, data collection and analysis, decision to publish, or preparation of the manuscript.

**Competing interests:** The authors have declared that no competing interests exist.

in highly invasive MDA-MB-231 stem cells was significantly higher than that of weakly invasive MCF-7 stem cells [7].

MicroRNAs, a class of non-coding RNA molecules of 19–25 nucleotides, are widely involved in the regulation of development, proliferation, differentiation and apoptosis [8–10]. Recently, microRNAs have rapidly developed into important molecular markers of tumors and other diseases [11, 12]. MiR-206 is a typical multifunctional microRNA that located in the non-coding gene Bic and expressed in a variety of cancer cells. It has been shown that miR-206 plays an important role in the process of immune response, inflammation, tumorigenesis and so on [13, 14]. There are also reported that the expression of miR-206 was down-regulated in estrogen receptor-alpha positive breast cancer MDA-MB-231 stem cells [15], and the overexpression of miR-206 in breast cancer MDA-MB-231 stem cells can effectively inhibit cells growth. Although these studies have shown that miR-206 plays an important role in the development and progression of breast cancer stem cells, it is still unclear how it regulates the growth of breast cancer stem cells.

In this study, we investigate the potential mechanism of miR-206 affecting the biological activity of breast cancer stem cells by interfering with the expression of CALR, and then providing novel targets and new strategy for the prevention and treatment of breast cancer.

## Materials and methods

### Chemicals and agents

miR-206 mimics, Ad-EVI-1, and Ad-CALR (Ambion,Tokyo, Japan); BCA Protein Assay Kit (Catalog number: 23225, Pierce Chemical, Rockford, USA); anti-EVI-1(463–477), anti-E-cadherin (SRP6426), anti-CALR (AV30114), anti-N-cadherin (05–915) were purchased from Sigma-Aldrich, USA; the Annexin V-FITC Apoptosis Detection Kit (Beyotime Institute of Biotechnology, Jiangsu, China); Total RNA was isolated from breast CSCs with RNAisoPlus (Takara Biotechnology Co, Ltd., Dalian, China). All cell culture medium and fetal bovine serum were purchased from Gibco (Thermo Fisher scientific, USA).

### Cell lines and cell culture

Human breast cancer cell line MDA-MB-231 was purchased from the Chinese Academy of Sciences Cell Bank (Shanghai, China). MDA-MB-231 cells were digested by trypsin and harvested. The cells were labeled with anti-human CD44-fluorescein isothiocyanate (FITC), anti-human CD24-phycoerythrin and anti-human ESA-PerCP-Cy5.5-A antibodies for 20 minutes. Breast cancer stem cells were sorted and cultured in DMEM/F12 (1:1) medium with 20 μg/L basic fibroblast growth factor, 10 μg/L epidermal growth factor and 2% B27, incubated at 37˚C, 5% $CO_2$.

### Transfection experiments

MiR-206 mimics, Ad-EVI-1 and Ad-CALR were transfected into human MDA-MB-231 breast cancer cells. After that, the cells were digested by trypsin and harvested. Labeled with anti-human CD44-fluorescein isothiocyanate (FITC), anti-human CD24-phycoerythrin and anti-human ESA-PerCP-Cy5.5-A antibodies for 20 minutes, breast cancer stem cells were sorted and cultured in DMEM/F12 (1:1) medium with 20 ug/L basic fibroblast growth factor, 10 ug/L epidermal growth factor and 2% B27, incubated at 37˚C, 5% $CO_2$. Breast cancer stem cells were inoculated into 6-well plates, cultured to 50% fusion, transfected with Lipofectamine 2000, and cultured in 10% fetal bovine serum for 24 hours. Then a new medium containing 100 U/mL penicillin and 100ng/mL streptomycin was replaced. Transfection efficiency was confirmed by quantitative RT-PCR.

## MTT assay

Breast cancer stem cells ($2 \times 10^3$) were seeded in 96-well plates, then transfected with 50 nM miR-206 and/or Ad-CALR constructs and incubated for 48 hours. The supernatant was removed, and an MTT solution was added to each well and cultured at 37˚C for additional 3 hours. The culture solution was aspirated, DMSO was added to each well and shaken for 10 min on a constant temperature shaker and incubated at room temperature for 20 min. The OD value was measured by microplate reader at 570 nm.

## Flow cytometry analysis

After transfection of 50 nM miR-206 or/and Ad-CALR for 48 hours, the cells were washed twice with cold PBS and resuspended in 1 x Binding Buffer. Take 195 μl of cell suspension in a 5 ml flow tube, add 5 μl of Annexin V-FITC, mix gently and mix for 3 min, then add 10μl or 20 μg/ml of PI solution, shake, and let stand for 15 min at room temperature in the dark. The tube was added with 400 μl of 1 x Binding Buffer. The apoptotic cells were astained and analyzed by flow cytometry.

## Transswell assay detects the invasion of breast cancer stem cells

Breast cancer stem cells were transfected with 50 nM miR-206 or/and Ad-CALR constructs and incubated for 48 hours. A single cell suspension was prepared and the cell concentration was adjusted to $2 \times 10^5$. Matrigel was used to cover the PVP membrane between the upper and lower rooms of Transswell and treated at 37˚C for 30 minutes. 100 μl of cells were added to the upper chamber, 500μl of serum-containing medium was added to the lower chamber. Each group was divided into three compound pores. After 24 hours of incubation in $CO_2$ incubator, the cells were removed and fixed with 95% ethanol for 10 minutes. Conventional 0.1% crystal violet staining, five visual fields were randomly taken under the microscope, up, down, left, right and center. The number of perforating cells was counted. The average number of each visual field was taken to indicate the invasiveness of tumor cells.

## Western blot analysis

As our previously described, equal amounts of protein were loaded onto an 8% SDS-PAGE gel and transferred to a polyvinylidene fluoride membrane.5% skimmed milk was sealed for 2 hours and then with anti-EVI-1 (1: 1,000), anti-CALR (1:500), anti-E-cadherin (1: 1,000) or anti-N-cadherin (1: 1,000) incubate together. After rinsed by TBST buffer, the membrane was incubated with a horseradish peroxidase-conjugated secondary antibody for 1 hour at a concentration of 1:5000. Finally, the density of immunoblot was quantified by image J software.

## Real-time quantitative PCR assay

Total RNA was isolated from breast cancer stem cells with RNA Plus according to the manufacturer's instructions. cDNA for microRNA and gene expression analyses was prepared using the miScript II RT kit and RT2 the first strand kit, respectively. The qRT-PCR was performed in triplicate using the miScript SYBR Green PCR kit and the Quantitech SYBR green kit on a StepOnePlus real time PCR system. Relative changes in expression were calculated using the $2^{-\Delta\Delta Ct}$ method, where $\Delta Ct$ is the difference in threshold cycles for the target gene and reference (ACTB), and $\Delta\Delta Ct$ is the difference between the $\Delta Ct$s of the treated sample and control or calibrator. Thus, the expression levels were reported as fold changes relative to the calibrator. The primers as follow: CALR- F 5′–TGG TCC TGG TCC TGA TGT CG–3′ and CALR-R 5′–

CTC TAC AGC TCG TCC TTG–3′; ACTN-F AGG TCA TCA CCA TCG GCA ACG A,
ACTN-R GCT GTT GTA GGT GGT CTC GTG A.

## Statistical methods

All data were analyzed by one-way ANOVA method using SPSS 13.0 software. $P < 0.05$ indicates that the difference has statistical significance.

## Results

### Effect of miR-206 on the expression of CALR in breast cancer stem cells

To investigate the effect of miR-206 on CALR expression in human breast cancer stem cell, the breast cancer stem cells were firstly sorted and collected as CD44+ and CD24+ positive population and then were transfected with 30, 50 and 70nM of miR-206 for 48 hours. As seen in Fig 1, miR-206 mimics significantly suppressed the protein and mRNA expression level of CALR in breast cancer stem cells in a dose-dependent manner. Treatment of miR-206 (50nM) resulted in approximately 50% reduction of CALR expression.

Furthermore, to determine the time-course inhibitory effect of miR-206 on CALR expression, human breast cancer stem cells were transfected with miR-206 mimics (50nM) at different time point (0, 24, 48 and 74 hours), and the mRNA and protein expression level of CALR was determined by RT-PCR and western blot analysis, respectively. As shown in Fig 1D, we observed that the significantly suppression effect on CALR expression in breast cancer stem cells when the cells were exposed to 50nM of miR-206 mimics for 48 or 72 hours. (Fig 1D–1F).

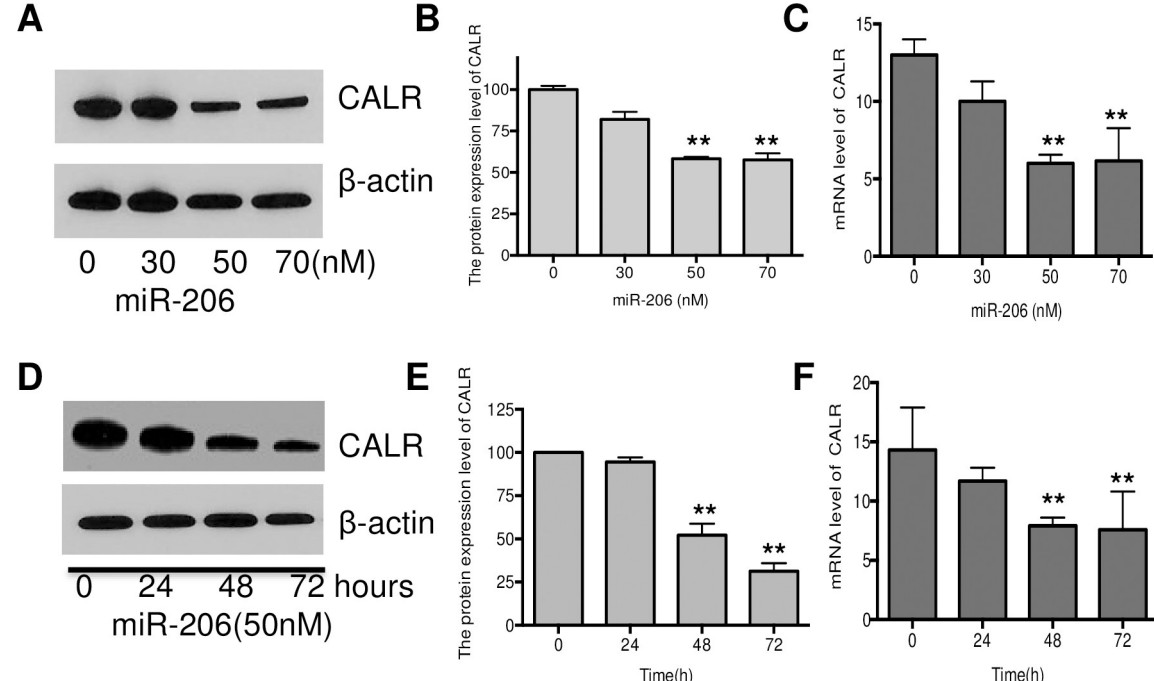

**Fig 1. Effect of miR-206 on CALR expression in breast cancer stem cells.** Human breast cancer stem cells were treated with various concentrations of miR-206 mimics (0, 30, 50, 70 nM) for 48 h. CALR protein expression (A&B) and mRNA expression (C) level were measured by Western blot and real-time quantitative PCR, respectively; Human breast cancer stem cells were treated with 50 nM miR-206 mimics for 0, 24, 48 and 72h. CALR protein expression (D&E) and mRNA expression level were determined by Western blot and real-time quantitative PCR, respectively. All results are expressed as the mean ± S.D. from three independent experiments. *$P < 0.05$, **$P < 0.01$ vs. control group.

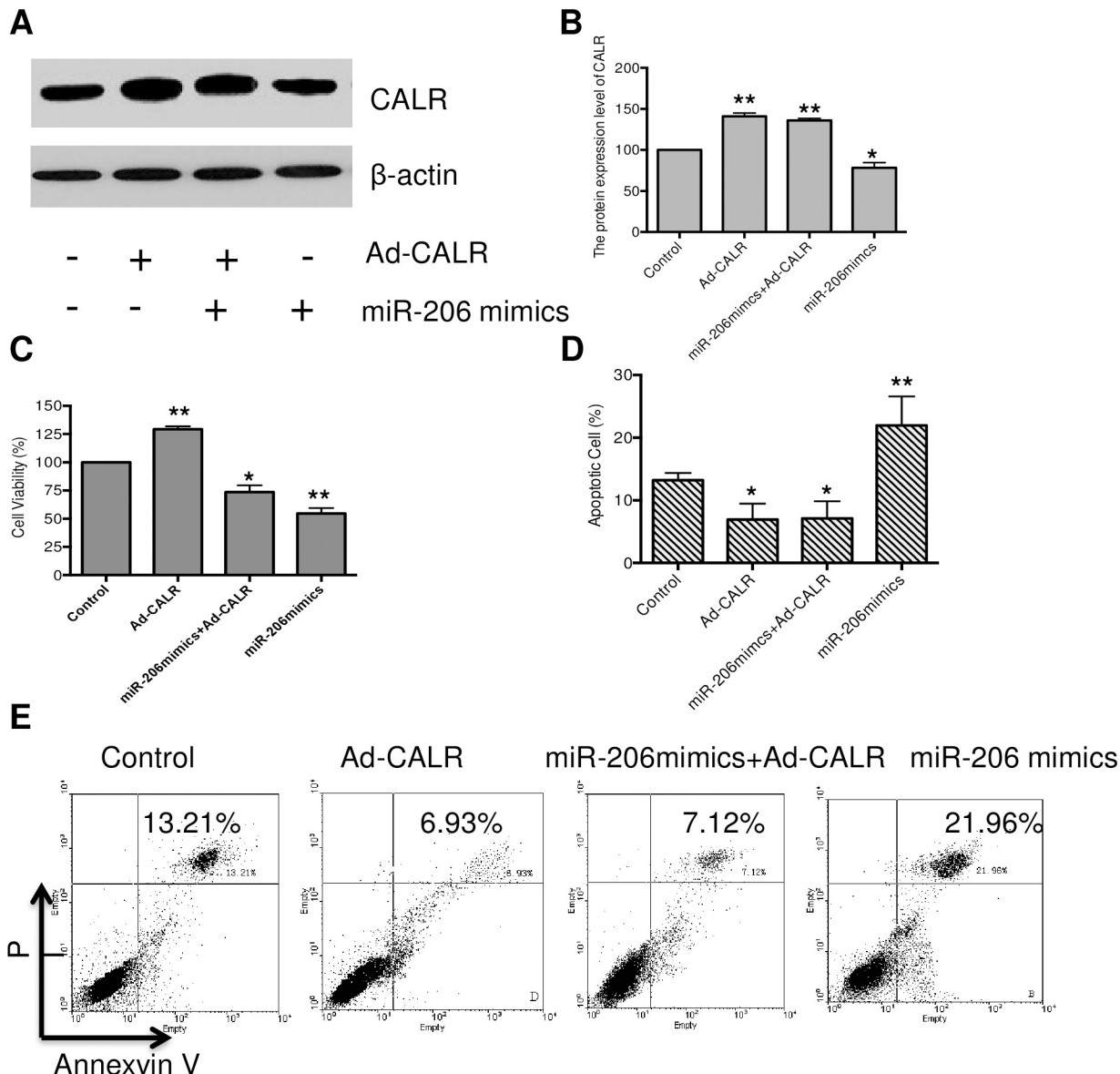

**Fig 2. Overexpression of CALR attenuate the cytotoxic effect of miR-206 on growth of breast cancer stem cells.** Human breast cancer stem cells were treated by miR-206 and/or Ad-CALR for 48 h, then the cells were harvested and analyzed. (A&B) CALR protein expression were determined by immunoblot analysis; (C) Human breast cancer stem cells were transfected with miR-206 and/or Ad-CALR for 72 h, the cell growth was determined by MTT assay; (D&E) Effect of CALR overexpression attenuate the apoptosis effect of miR-206 in breast cancer stem cells. Human breast cancer stem cells were treated by miR-206 and/or Ad-CALR for 48 h, the early and late apoptotic cells were measured by PI-Annexin V, $^*P < 0.05$, $^{**}P < 0.01$ vs. control group.

### Overexpression of CALR attenuate the cytotoxic effect of miR-206 in breast cancer stem cells

To verify that CALR can be acted as the target of miR-206, the ad-CALR was designed and transfected into breast cancer stem cells for 48 hours. As shown in Fig 2A & 2B, the expression of CALR was significantly elevated in ad-CALR alone group, and miR-206 alone significantly reduced CALR expression. While the co-transfection of ad-CALR and miR-206, the overexpression of CALR could attenuate the inhibitory effect of miR-206 on CALR expression (P

<0.05). The effect of ad-CALR and/or miR206 on the growth of breast cancer stem cells was determined by MTT assay. As seen in Fig 2C, miR-206 mimics alone treatment for 48 hours resulted in 52.96% suppressing cell growth, which significant inhibited the proliferation of breast cancer stem cells as compared to the control group (P<0.05). Meanwhile, we observed that the transfection with Ad-CALR promoted the growth of cells. The breast cancer stem cells were transfected with Ad-CALR in combination with miR-206 mimics for 48 hours, the inhibitory effect of miR-206 was reversed by overexpression of CALR.

The results showed that miR-206 can significantly inhibit the growth of breast cancer stem cells, but the effect of miR-206 on the growth of breast cancer stem cells was significantly attenuated by overexpression of CALR (Fig 2C). In addition to the MTT assay, the effect of miR-206 on inducing cellular apoptosis was measured by PI-AnnexinV staining assay. As seen in Fig 2D & 2E, the cells were treated by miR-206 alone for 48 hours, the late apoptotic cells were increased to 21.96%. Meanwhile, we found that overexpression of CALR reduced the cells undergoing apoptosis (6.96%). What is more, the apoptotic cells induced by miR206 were significantly deceased when the cells were co-transfected with Ad-CALR. Thus, overexpression of CALR could effectively attenuate the cytotoxic effect of miR-206 in breast cancer stem cells.

## Effect of miR-206 on CALR-mediated metastasis of human breast cancer stem cells

Next, the effect of miR-206 mimics and/or ad-CALR on the ability of invasion and migration was measured by using transwell assay. As seen in Fig 3A, the breast cancer stem cells were transfected with miR206 alone for 48 hours, the invasive cells were decreased 31.9%, as compared to the control group (P<0.05). Overexpression of CALR enhanced the ability of breast cancer stem cells, and the invasive cells were increased to 127.7%, (p<0.05). What is more, the co-transfection of Ad-CALR significantly reduced the suppression of miR206 on invasive breast cancer stem cells. Our results suggested that miR-206 could significantly inhibit the invasion ability of breast cancer stem cells, whereas the ability of miR-206 to inhibit the invasion of breast cancer stem cells is reduced by overexpression of CALR. (Fig 3).

EMT (epithelial-mesenchymal transition) is essential for initiation of metastasis in tumor progression. Given the effect of miR206/CALR on cell invasion, expression level of key regulators in EMT signaling pathway was determined by western blot analysis, including E-cadherin and N-cadherin. As seen in Fig 3B, overexpression of CALR enhanced the E-cadherin and reduced N-cadherin expression. miR-206 alone resulted in a decrease of E-cadherin and increase of N-cadherin. Co-transfection of Ad-CALR could partially reduce the biological effect of miR206 on EMT pathway. This result is consistent to our transwell assay. In addition, our results provide the evidences that miR-206 can regulate the expression of EMT-related genes by enhancing the expression of CALR.

## EVI-1 is associated with miR206/CALR signaling in human breast cancer

To clarify the mechanism of miR-206 on CALR expression, the effect of EVI-1 on CALR expression was investigated. As shown in Fig 4A & 4B, CALR expression in breast cancer stem cells was significantly increased in Ad-EVI-1 group, as compared to the control group; while knockdown of EVI-1 (si-EVI-1) effectively reduced CALR expression in breast cancer stem cells (P<0.05). There is a positive correlation of CALR and EVI-1 in breast cancer stem cells.

To further confirm the function of EVI-1 in miR206-mediated CALR expression, the effect of Ad-EVI and/or miR-206 on EVI-1 and CALR expression were determined by western blot analysis and RT-PCR, respectively. As the results shown in Fig 4C & 4D, EVI-1 and CALR expression in breast cancer stem cells were significantly suppressed in the miR-206 mimics

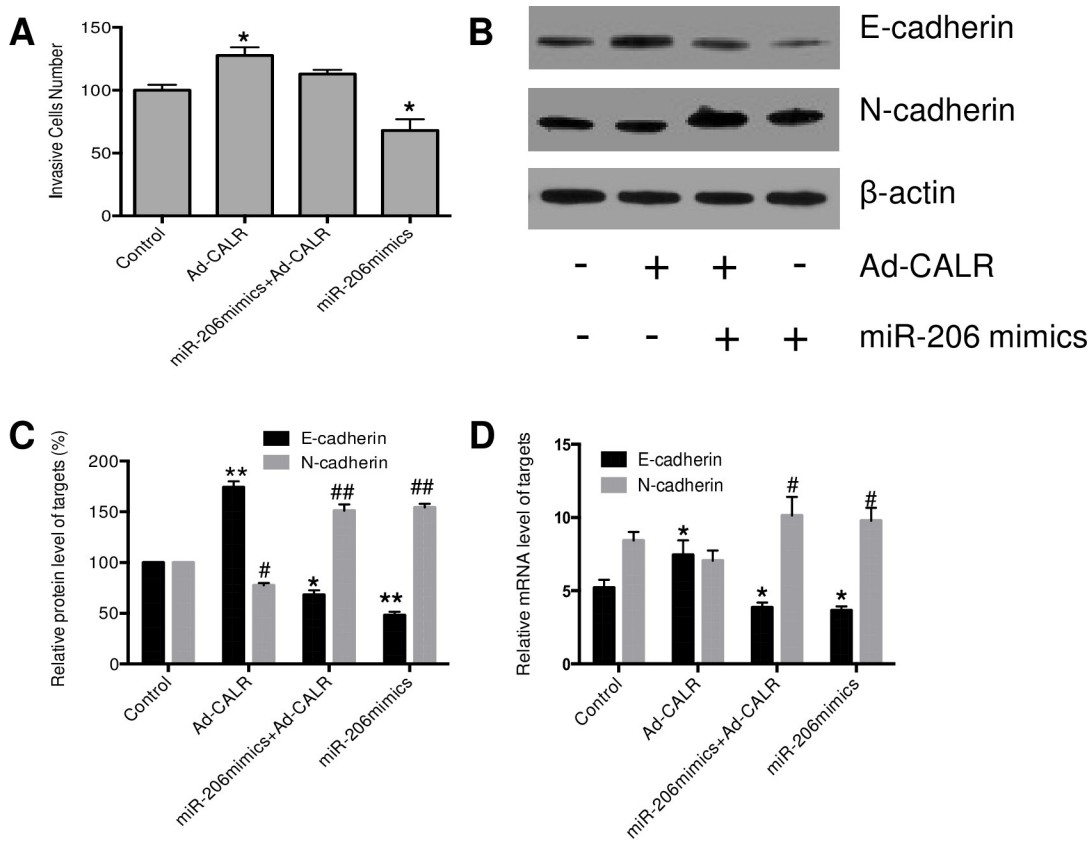

**Fig 3. Effect of miR-206 on CALR-mediated metastasis of human breast cancer stem cells.** Cells were treated with miR-206 mimics, miR-206 mimics+Ad-CALR and Ad-CALR for 48 h, respectively. (A) invasive breast cancer stem cells were counted; (B&D), CALR protein expression was measured by Western blotting analysis; C, CALR gene was measured by real-time quantitative PCR. All results are expressed as the mean ± S.D. from three independent experiments, $^*P < 0.05$, $^{**}P < 0.01$; #$P<0.05$, ##$P<0.01$ vs. control group.

group, as compared to the control group (P<0.05). The expressions of EVI-1 and CALR were significantly enhanced in miR-206 mimics plus Ad-EVI-1 group and Ad-EVI-1 alone group (P<0.05). These results indicated that the inhibitory effect of miR-206 on the expression of CALR may be reversed the overexpression of EVI-1.

## Discussion

miR-206 has been detected in various types of tumor cells, including lung cancer, gastric cancer, breast cancer and colon cancer. Functionally, MiR-206 can influence fate of tumor cells by regulating the expression levels of different signaling pathways in tumor cells, and then manipulate the formation of tumors [16, 17]. Breast cancer stem cells not only retain the basic characteristics of normal cells, but also have strong self-renewal ability and tumorigenicity. This feature may be one of the main reason of recurrence and metastasis of breast cancer, but the effect and precise mechanism of miR206 on proliferation and metastasis of breast cancer stem cells is unknown.

The expression level of CALR is positively correlated with tumor size and status of metastasis. Thus, CALR can be used as an auxiliary biomarker to guide diagnosis, treatment and prognosis [18]. Herein, we discovered that miR-206 mimics inhibited CALR expression in a time- and dose-dependent manner. RT-PCR results confirmed that miR-206 mimics significantly

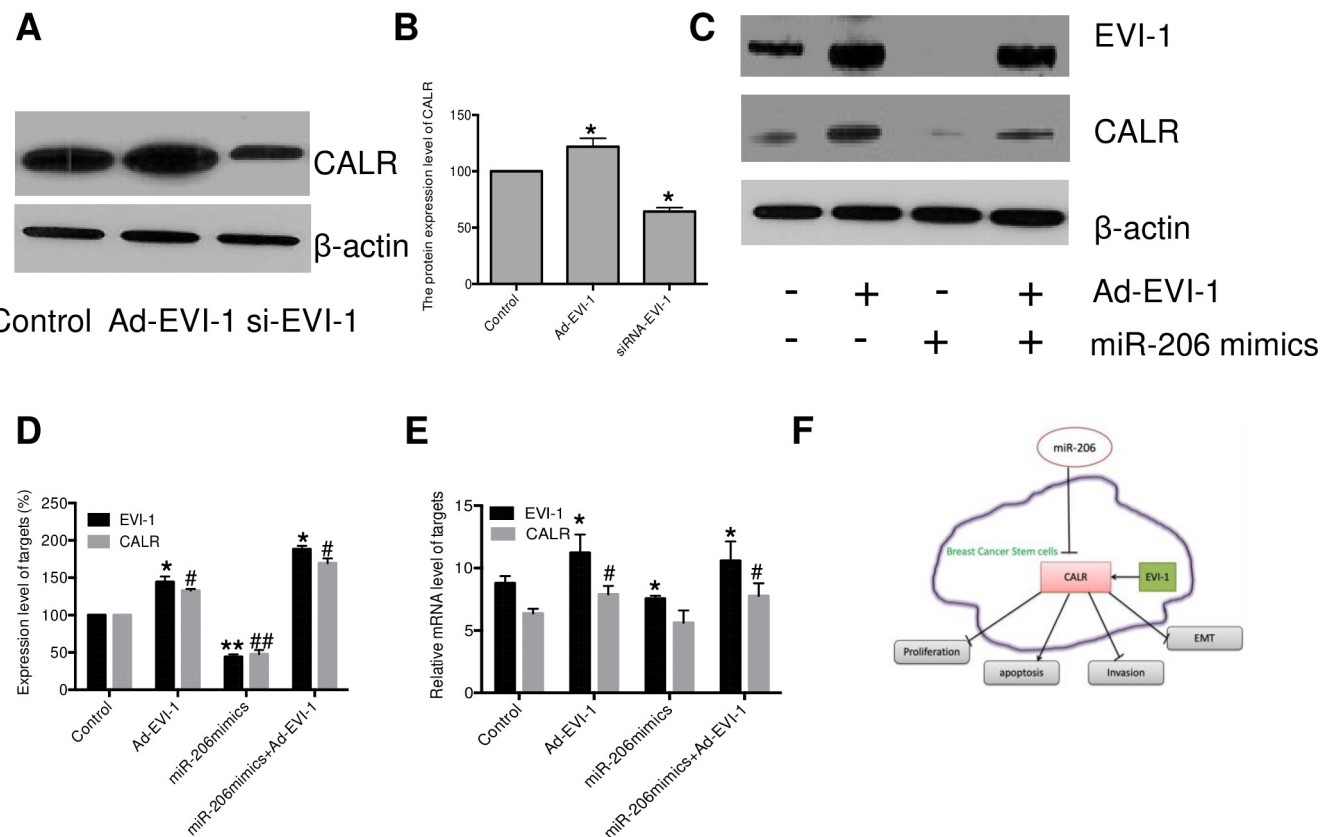

**Fig 4. EVI-1 is involved in miR-206/CALR in human breast cancer stem cells.** (A&B) breast cancer stem cells were transfected with Ad-EVI-1 or si-EVI-1 for 48 h, the EVI-1 expression was determined by immunoblot analysis; (C&D) Human breast stem cell were treated with miR-206 mimics, miR-206 mimics +Ad-EVI-1 and Ad-EVI-1 for 48 h. EVI-1 and CALR expression were measured by Western blotting analysis; (E), the mRNA level of EVI-1 and CALR were measured by real-time quantitative PCR; (F), Schematic of MiR/206-EVI-1/CALR pathway in human breast cancer. All results are expressed as the mean ± S. D. from three independent experiments, $^*P < 0.05$, $^{**}P < 0.01$; #P<0.05, ##P<0.01 vs. control group.

suppressed CALR mRNA. Our results suggest that enhancing the expression of miR-206 can significantly inhibit CALR expression in breast cancer stem cells.

Further studies demonstrated that miR-206 mimics significantly induced apoptosis and suppressed the proliferation and invasion ability of breast stem cells. Interestingly, overexpression of CALR could attenuate the effect of miR-206 mimics. However, there was no significant difference between the Ad-CALR alone and control group. This result suggested that miR-206 play an important role in growth of breast cancer stem cells, and closely related to CALR expression.

EMT play an important role in the invasion and metastasis of malignant tumors, and main change in EMT in tumor cells is a decrease in the expression of the epithelial marker E-cadherin and an increase in the expression of the interstitial marker N-cadherin [19–21]. Overexpression of N-cadherin is closely related to the invasion and metastasis of a variety of epithelial-derived malignant tumors. When N-cadherin is over-expressed, the structure of tissue cells becomes loose, the adhesion between cells decreases, and then cells Invasion and transfer [22, 23]. Herein, we observed that miR-206 could significantly enhance the expression of E-cadherin and decrease the expression of N-cadherin. What is more, this effect was reversed when the expression of CALR was upregulated. Our results indicated that miR-206 altered regulation of E-cadherin and N-cadherin through CALR-dependent manner in breast cancer stem cells.

Recently, researchers found that the expression of EVI-1 (ecotropic virus integration-1) is up-regulated in metastatic breast cancer stem cells. High expression of EVI-1 can lead to poor prognosis in breast cancer patients with estrogen receptor negative [24–26], suggesting that transcription factor EVI-1 may play a carcinogenic role by promoting the growth, migration and invasion of breast cancer stem cells. Our study demonstrated that enhancing the expression of EVI-1 could significantly enhance the expression of CALR in breast cancer stem cells, and there was a significant positive correlation between CALR and EVI-1. Further study indicated that high-level of miR-206 could reduce the expression of EVI-1 and inhibit the activity of CALR, but when the expression of EVI-1 was enhanced, the effect of miR-206 on inhibiting CALR activity was significantly reduced. These results suggested that miR-206 was likely to regulate the activity of CALR by reducing the expression of EVI-1.

Taken together, miR-206 suppresses the growth and metastasis ability of breast cancer stem cells via blocking CALR expression. Further studies demonstrated that regulation of CALR by miR-206 may be realized by affecting the expression of EVI-1. Our result not only clarified the mechanism of miR-206 regulating CALR expression in breast cancer stem cells, but also provided a novel target for the prevention and treatment of breast cancer.

## Supporting information

**S1 File. Raw images for western blots.**
(PDF)

## Acknowledgments

We gratefully thank the staff in the Department of Oncology, The First Affiliated Hospital of Jinzhou Medical University who reviewed and made comments on our manuscript.

## Author Contributions

**Conceptualization:** Dapeng Sun, Fengxiang Zhang.

**Data curation:** Fengxiang Zhang.

**Funding acquisition:** Fengxiang Zhang.

**Methodology:** Chenguang Li, Fengxiang Zhang.

**Writing – original draft:** Dapeng Sun, Chenguang Li, Fengxiang Zhang.

**Writing – review & editing:** Fengxiang Zhang.

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
