## [Decision Letter · Decision Letter 0]

31 May 2022

PONE-D-22-08405MicroRNA-206 suppresses growth and metastasis of breast cancer stem cells via blocking EVI-1-mediated CALR expressionPLOS ONE

Dear Dr. zhang,

Thank you for submitting your manuscript to PLOS ONE. After careful consideration, we feel that it has merit but does not fully meet PLOS ONE’s publication criteria as it currently stands. Therefore, we invite you to submit a revised version of the manuscript that addresses the points raised during the review process. According to all reviewers' comments, please address the concerns. The quality of this manuscript will be improved.

We look forward to receiving your revised manuscript.

Kind regards,

Ning Wei

Academic Editor

PLOS ONE

Journal Requirements:

“This study was supported by Liaoning Provincial Natural Science 11 Foundation (20180550601); Liaoning Provincial Natural Science Foundation(2019- ZD-0833).”

Additional Editor Comments:

According to all reviewers' comments, please address the concerns. The quality of this manuscript will be improved.

Reviewers' comments:

Reviewer's Responses to Questions

**Comments to the Author**

1. Is the manuscript technically sound, and do the data support the conclusions?

Reviewer #1: Partly

Reviewer #2: Yes

Reviewer #3: Yes

2. Has the statistical analysis been performed appropriately and rigorously? 

Reviewer #1: No

Reviewer #2: Yes

Reviewer #3: Yes

3. Have the authors made all data underlying the findings in their manuscript fully available?

Reviewer #1: Yes

Reviewer #2: Yes

Reviewer #3: Yes

4. Is the manuscript presented in an intelligible fashion and written in standard English?

Reviewer #1: No

Reviewer #2: Yes

Reviewer #3: Yes

5. Review Comments to the Author

Reviewer #1: In this paper, the authors found that miR-206 mimics suppressed CALR expression, inhibited the proliferation and metastasis ability of breast cancer stem cells and finally induced cellular apoptosis. They further showed that this was correlated with suppression of EVI-1 signaling. The study was supported some solid data. However, few issues need to be addressed.

1. Please clarify how or where did you get the stem cells?

2. The materials and methods part is completely in a mess, please rewrite this part clearly.

3. In Fig 1B, how did the authors qualify the protein expression, please clarify in the methods part.

4. As showed in Fig 1D and E, the suppression effect was stronger at 72 hours instead of 48 hours, please check this result.

5. In Fig 2C and D, why the MTT assay did not inconsistent with flow analysis in Ad-CALR and Ad-CALR + miR-206 mimics group?

6. Could the authors provide representative images of transwell assay?

7. Many grammar mistakes and errors were noted. It is suggested that authors seek for professional editing service to improve the quality of language and the organization of the manuscript.

Reviewer #2: On the whole, it is a valuable and meaningful study, and the design in this study is logical. However, some points should be explained as following before publication.

1. Its research significance and potential application value should be discussed in the discussion

2. How to prove whether the transfection of MiR-206 mimics is successful?

3. What is the function of E-cadherin and N-cadherin in tumor progression? Why CALR exerted different effect on E-cadherin and N-cadherin?

4. Why Ad-CALR can not reverse the effect of MiR-206 on E-cadherin and N-cadherin?

5. Did the Ad-EVI-1 treatment affect the effect of miR-206 mimics on cytotoxic effect, E-cadherin and N-cadherin?

6. Please explained the principle and mechanism of miR-206 mimics, Ad-CALR and Ad-EVI-1 transfected into cells, what is different with plasmid transfection?

Reviewer #3: In this manuscript, Dapeng and co-authors describe the effect of miRNA-206 on the growth and migration of breast cancer cells and explore the underlying mechanism. Defining the role of miRNA-206 in breast cancer progression is useful to develop novel relevant prevention and/or therapy strategy . The authors did some work to validate their proposal. However, this study is not logically clear and some data is not convincing and need more detail work relating the underlying mechanism, particularly in relation to the relationship between miR-206, CALR and EVI-1 in their engaged signaling pathway.

Major comments

1. MM doesn’t clarify the origin of breast cancer stem cell, and more importantly these used stem cells should be fully/extensively characterized prior to functional assay, such as more stem cell markers besides CD44+/ CD24- should be employed to validate stem cells identity.

2. For the stem cells invasion/migration assay, it’s worth to include some representative images, that will be more visualized and convincing.

3. The final schematic is not consistent with the conclusion drawn in the text.

4. It is highly recommended to carefully go over the MM part to sort out some obvious incorrect description such as “transfected with 5ml Lipo 200”, repeated description, and incomplete method description.

5. Please include all the qPCR primers sequence used in this study.

6. The EVI-1 background introduction should be incorporated.

Minor comments

a. The cat # of Reagents/Cells/Instrument used in this study is missing.

b. Please number all the rows of the text part, which will be much easier to be located during the reviewing process.

c. The ms should be further polished regarding its writing in a professional way and tense using.

6. PLOS authors have the option to publish the peer review history of their article (what does this mean?). If published, this will include your full peer review and any attached files.

Reviewer #1: No

Reviewer #2: No

Reviewer #3: No

---

## [Author Response · Author response to Decision Letter 0]

27 Aug 2022

Response: Dear Editor, according to your suggestion, the title page was updated.

Response: Dear Editor, according to your suggestion, the funding information was added on page 10.

“This study was supported by Liaoning Provincial Natural Science 11 Foundation (20180550601); Liaoning Provincial Natural Science Foundation(2019- ZD-0833).”

Response: Dear Editor, according to your suggestion, the acknowledgement was corrected.

Response: Dear Editor, thank you for your suggestion! The supporting information files were uploaded, include the western blot images.

Response: Dear Editor, according to you comment, the ORCID (0000-0002-1713-4035) was provided.

Response: Dear Editor, the western blot images were uploaded.

Point-to-point Response to reviewer

Reviewer #1: In this paper, the authors found that miR-206 mimics suppressed CALR expression, inhibited the proliferation and metastasis ability of breast cancer stem cells and finally induced cellular apoptosis. They further showed that this was correlated with suppression of EVI-1 signaling. The study was supported some solid data. However, few issues need to be addressed.

1. Please clarify how or where did you get the stem cells?

Response: Dear Reviewer#1, thank you for your concern. As described in the method 2.2, MDA-MB-231 cells were digested by trypsin and harvested. The cells were labeled with anti-human CD44-fluorescein isothiocyanate (FITC), anti-human CD24-phycoerythrin and anti-human ESA-PerCP-Cy5.5-A antibodies for 20 minutes. Breast cancer stem cells were cultured in DMEM/F12 (1:1) medium with 20 µg/L basic fibroblast growth factor, 10 µg/L epidermal growth factor and 2% B27, incubated at 37°C, 5% CO2. (on page 4, line 112).

2. The materials and methods part is completely in a mess, please rewrite this part clearly.

Response: Reviewer#1, thank you for your suggestion! As seen on page3 line 94, We carefully revised and edited the part of method and made it clear. 

3. In Fig 1B, how did the authors qualify the protein expression, please clarify in the methods part.

Response: thank you for your suggestion! The quantification of protein expression was performed by image J, the sentence was inserted into the part of method (on page 5, line 180).

4. As showed in Fig 1D and E, the suppression effect was stronger at 72 hours instead of 48 hours, please check this result.

Response: Thank you for your comment! The effect of miRNA206 on CALR expression was quantified by image J software, and an updated image was shown in Fig.1D. As seen, the stronger effect of miR-206 was observed at 72 hours. 

5. In Fig 2C and D, why the MTT assay did not inconsistent with flow analysis in Ad-CALR and Ad-CALR + miR-206 mimics group?

Response: Dear Reviewer#1, we have double checked the results, the cell viability means the live breast cancer stem cells, and the apoptotic cells means the dead cells. The ad-CALR promote cell growth, and miR-206 induce cellular apoptosis. The detail of this description was showed on page 7, line 206.

6. Could the authors provide representative images of transwell assay?

Response: Dear Reviewer#1, thank you for your concern. We did not take image for the transwell assay, and only counted the metastatic cells under the microscope.

7. Many grammar mistakes and errors were noted. It is suggested that authors seek for professional editing service to improve the quality of language and the organization of the manuscript.

Response: Dear Reviewer#1, thanks for your comment! We have carefully re-written and checked the grammar mistakes and errors in the full-text. We also asked an English native speaker for the correction.

Reviewer #2: On the whole, it is a valuable and meaningful study, and the design in this study is logical. However, some points should be explained as following before publication.

1. Its research significance and potential application value should be discussed in the discussion.

Response: Dear Reviewer#2, thank you for your suggestion! Some sentences on research significance and potential application value were added on the part of discussion. For example: “Our result not only clarified the mechanism of miR-206 regulating CALR expression in breast cancer stem cells, but also provided a novel target for the prevention and treatment of breast cancer. ” on page 10.

2. How to prove whether the transfection of MiR-206 mimics is successful?

Response: Dear Reviewer#2, thank you for your comment. Previously, the transfection of microRNA has been performed in our lab. The transfection rate is over 85%. We confirmed that our transfection method worked in this study.

3. What is the function of E-cadherin and N-cadherin in tumor progression? Why CALR exerted different effect on E-cadherin and N-cadherin?

Response: Dear Reviewer#2, as the description in the part of discussion on page 9 line 131, “EMT play an important role in the invasion and metastasis of malignant tumors, and main change in EMT in tumor cells is a decrease in the expression of the epithelial marker E-cadherin and an increase in the expression of the interstitial marker N-cadherin[19-21]. Overexpression of N-cadherin is closely related to the invasion and metastasis of a variety of epithelial-derived malignant tumors. When N-cadherin is over-expressed, the structure of tissue cells becomes loose, the adhesion between cells decreases, and then cells Invasion and transfer[22, 23].”

4. Why Ad-CALR can not reverse the effect of MiR-206 on E-cadherin and N-cadherin?

Response: Dear Reviewer#2, thank you for your concern! As we described in the part of result on page 8 line 219, “As seen in Fig.3B, overexpression of CALR enhanced the E-cadherin and reduced N-cadherin expression. miR-206 alone resulted in a decrease of E-cadherin and increase of N-cadherin. Co-transfection of Ad-CALR could partially reduce the biological effect of miR206 on EMT pathway. This result is consistent to our transwell assay”.

5. Did the Ad-EVI-1 treatment affect the effect of miR-206 mimics on cytotoxic effect, E-cadherin and N-cadherin?

Response: Dear Reviewer#2, in this study, we did investigate the effect of Ad-EVI-1 on miR-206 on cytotoxic effect and E-cadherin, N-cadherin expression. In fact, we only focused on the regulation of CALR expression in current study. In the future, we will investigate the function of EVI-1 in breast cancer stem cells.

6. Please explained the principle and mechanism of miR-206 mimics, Ad-CALR and Ad-EVI-1 transfected into cells, what is different with plasmid transfection?

Response: Dear Reviewer#2, thank you for your comment! Based on our result, the dose of miR-206 was chosen as 50nM. The amounted of Ad-CALR and Ad-EVI, we tested a series of doses, and the amount of 500ug of this plasmid could effectively overexpression the target protein. 1000 ug of plasmid is too toxic. Thus, 500 ug of plasmid was choose. 

Reviewer #3: In this manuscript, Dapeng and co-authors describe the effect of miRNA-206 on the growth and migration of breast cancer cells and explore the underlying mechanism. Defining the role of miRNA-206 in breast cancer progression is useful to develop novel relevant prevention and/or therapy strategy . The authors did some work to validate their proposal. However, this study is not logically clear and some data is not convincing and need more detail work relating the underlying mechanism, particularly in relation to the relationship between miR-206, CALR and EVI-1 in their engaged signaling pathway.

Major comments

1. MM doesn’t clarify the origin of breast cancer stem cell, and more importantly these used stem cells should be fully/extensively characterized prior to functional assay, such as more stem cell markers besides CD44+/ CD24- should be employed to validate stem cells identity.

Response: Dear Reviewer#3, thank you for your suggestion! In our study, the breast cancer stem cells were stained and sorted by flow cytometry. We will investigate the function of breast cancer stem cell in the future study.

2. For the stem cells invasion/migration assay, it’s worth to include some representative images, that will be more visualized and convincing.

Response: Dear Reviewer#3, thank you for your comment! We did not take image for the transwell, and only count the cells under the microscope.

3. The final schematic is not consistent with the conclusion drawn in the text.

Response: Dear Reviewer#3, based on you comment! We have revised the schematic figure (as shown in Fig. 4F).

4. It is highly recommended to carefully go over the MM part to sort out some obvious incorrect description such as “transfected with 5ml Lipo 200”, repeated description, and incomplete method description.

Response: Dear Reviewer#3, thank you for your suggestion! We have carefully revised and edited the part of method.

5. Please include all the qPCR primers sequence used in this study.

Response: Dear Reviewer#3, according to your suggestion! The list of qPCR primers was listed in the part of method. As shown on page 6, line 193, “The primers as follow: CALR- F 5′-TGG TCC TGG TCC TGA TGT CG-3′ and CALR-R 5′-CTC TAC AGC TCG TCC TTG-3′; ACTN-F AGG TCA TCA CCA TCG GCA ACG A, ACTN-R GCT GTT GTA GGT GGT CTC GTG A.”

6. The EVI-1 background introduction should be incorporated.

Response: Dear Reviewer#3, based on your suggestion, the background of EVI-1 was inserted.

Minor comments

a. The cat # of Reagents/Cells/Instrument used in this study is missing.

Response: Dear Reviewer#3, thank you for your comment! The CatLog number of the agent was added in the part of method.

b. Please number all the rows of the text part, which will be much easier to be located during the reviewing process.

Response: Dear Reviewer#3, thank you for your suggestion! The row number was added in the revised manuscript.

c. The ms should be further polished regarding its writing in a professional way and tense using.

Response: Dear Reviewer#3, thanks for your comment! We have carefully re-written and checked the grammar mistakes and errors in the full-text. We also asked an English native speaker for the correction.

---

## [Decision Letter · Decision Letter 1]

7 Sep 2022

MicroRNA-206 suppresses growth and metastasis of breast cancer stem cells via blocking EVI-1-mediated CALR expression

PONE-D-22-08405R1

Dear Dr. Zhang,

We’re pleased to inform you that your manuscript has been judged scientifically suitable for publication and will be formally accepted for publication once it meets all outstanding technical requirements.

Kind regards,

Ning Wei

Academic Editor

PLOS ONE

Additional Editor Comments (optional):

The current version of this manuscript is acceptable.

Reviewers' comments:

Reviewer's Responses to Questions

**Comments to the Author**

1. If the authors have adequately addressed your comments raised in a previous round of review and you feel that this manuscript is now acceptable for publication, you may indicate that here to bypass the “Comments to the Author” section, enter your conflict of interest statement in the “Confidential to Editor” section, and submit your "Accept" recommendation.

Reviewer #1: All comments have been addressed

Reviewer #2: (No Response)

Reviewer #3: All comments have been addressed

2. Is the manuscript technically sound, and do the data support the conclusions?

Reviewer #1: Yes

Reviewer #2: (No Response)

Reviewer #3: Yes

3. Has the statistical analysis been performed appropriately and rigorously? 

Reviewer #1: Yes

Reviewer #2: (No Response)

Reviewer #3: Yes

4. Have the authors made all data underlying the findings in their manuscript fully available?

Reviewer #1: Yes

Reviewer #2: (No Response)

Reviewer #3: Yes

5. Is the manuscript presented in an intelligible fashion and written in standard English?

Reviewer #1: Yes

Reviewer #2: (No Response)

Reviewer #3: Yes

6. Review Comments to the Author

Reviewer #1: In this paper, the authors found that miR-206 mimics suppressed CALR expression, inhibited the proliferation and metastasis ability of breast cancer stem cells and finally induced cellular apoptosis. They further showed that this was correlated with suppression of EVI-1 signaling. The study was supported some solid data. And only a few issues need to be addressed.

1. Line 154-155, it should be actin instead of ACTN.

2. Please correct the uppercases and lowercases in the article.

3. For the Annexin V/PI staining, the authors can include Annexin V+ only part in Fig. E2.

Reviewer #2: (No Response)

Reviewer #3: Even the current version of the manuscript looks pretty decent, I would still strongly suggest performing extensive/full characterization of the stem cells in future study.

7. PLOS authors have the option to publish the peer review history of their article (what does this mean?). If published, this will include your full peer review and any attached files.

Reviewer #1: No

Reviewer #2: No

Reviewer #3: No

---

## [Editor Report · Acceptance letter]

12 Sep 2022

PONE-D-22-08405R1 

MicroRNA-206 suppresses growth and metastasis of breast cancer stem cells via blocking EVI-1-mediated CALR expression 

Dear Dr. Zhang:

I'm pleased to inform you that your manuscript has been deemed suitable for publication in PLOS ONE. Congratulations! Your manuscript is now with our production department. 

Kind regards, 

on behalf of

Dr. Ning Wei 

Academic Editor

PLOS ONE